# Observation of strong excitonic magneto-chiral anisotropy in twisted bilayer van der Waals crystals

Shoufeng Lan[1,2], Xiaoze Liu[1], Siqi Wang[1], Hanyu Zhu[1], Yawen Liu[3], Cheng Gong[1], Sui Yang [1], Jing Shi[3], Yuan Wang[1] & Xiang Zhang [1,4✉]

The interplay between chirality and magnetism generates a distinct physical process, the magneto-chiral effect, which enables one to develop functionalities that cannot be achieved solely by any of the two. Such a process is universal with the breaking of parity-inversion and time-reversal symmetry simultaneously. However, the magneto-chiral effect observed so far is weak when the matter responds to photons, electrons, or phonons. Here we report the first observation of strong magneto-chiral response to excitons in a twisted bilayer tungsten disulfide with the amplitude of excitonic magneto-chiral (ExMCh) anisotropy reaches a value of ~4%. We further found the ExMCh anisotropy features with a spectral splitting of ~7 nm, precisely the full-width at half maximum of the excitonic chirality spectrum. Without an externally applied strong magnetic field, the observed ExMCh effect with a spontaneous magnetic moment from the ferromagnetic substrate of thulium iron garnet at room temperature is favorable for device applications. The unique ExMCh processes provide a new pathway to actively control magneto-chiral applications in photochemical reactions, asymmetric synthesis, and drug delivery.

[1] Nanoscale Science and Engineering Center, University of California, Berkeley, CA, USA. [2] Department of Mechanical Engineering, Texas A&M University, College Station, TX, USA. [3] Department of Physics and Astronomy, University of California, Riverside, CA, USA. [4] Faculties of Sciences and Engineering, University of Hong Kong, Hong Kong SAR, China. ✉email: xzhang@me.berkeley.edu

The magneto-chiral (MCh) effect is a fundamental physical process that merges chirality and magnetism in a chiral medium[1–4]. Such a link has been explored ever since the discovery of the natural optical activity by Arago and magnetically induced optical activity by Faraday in the early 19th century[5]. Besides the observation in synthetic molecular compounds under a magnetic field[6–9], researchers have observed the MCh effect in organic liquids[10], diamagnetic crystals[11], magnetic thin films[12–14], liquid crystals[15], and chiral metamaterials[16]. The MCh effect also gives rise to an unambiguous enantiomeric excess in a photochemical reaction and a separation of racemic mixtures which were not possible solely by natural or magnetically induced optical activity[17,18]. However, the MCh effect observed so far is weak, especially in the visible spectral region at room temperature, with the MCh anisotropy on the level of 0.1%. Such a low MCh anisotropy that signifies the ability to induce the enantiomeric excess that is important for photochemical reactions, asymmetric synthesizes, and drug delivery thereby hinders many applications.

We discover the strong excitonic magneto-chiral (ExMCh) response of 4% in an atomically thin twisted bilayer tungsten disulfide (TB-WS$_2$), a transition metal dichalcogenide (TMD) crystal. We understand this strong effect with the unique excitonic processes associated with valley excitons in the TMD of our device. First, the valley excitons enhance the chirality in TB-WS$_2$ by strong spin–orbit coupling[19]. The valley excitons also facilitate a strong response to a magnetic field that lifts the degeneracy between the two valleys[20–24]. The interplay among valley excitons, chirality, and magnetic field generates a new physical process called the ExMCh effect. We further enhance the ExMCh effect via exchange magnetic interactions by stacking the atomically thin TB-WS$_2$ on top of a ferromagnetic substrate, thulium iron garnet (TIG), whose Curie temperature is higher than the room temperature[25,26]. Without an externally applied strong magnetic field, the observed strong ExMCh effect with the spontaneous magnetic moment from the TIG substrate at room temperature is desired for device applications and hence offering new opportunities to manipulate MCh systems.

## Results

In such systems, light-matter interactions behave quantum mechanically with the Hamiltonian written as $H_{interaction} = -\mu \cdot \mathbf{E} - m \cdot \mathbf{B}$, where $\mu$, $m$, $\mathbf{E}$, and $\mathbf{B}$ are electric dipole, magnetic dipole, electric field, and magnetic field, respectively. For a common medium, the interaction between electric dipoles ($\mu\mu$), such as the ubiquitous Rayleigh scattering, that is even regarding both the parity-inversion and time-reversal symmetry (Fig. 1a) dominates. For a chiral medium, on the other hand, the parity-inversion symmetry breaking induces a magnetic dipole interaction. Thus, the chiral light-matter interaction (Fig. 1b) that keeps the time-reversal symmetry conserved is a product of $im\mu$, where $i$ is a result of the time-reversal symmetry operator in quantum mechanics. Moreover, when applied a static magnetic field that breaks the time-reversal symmetry, light-matter interactions in a common medium also contain a magnetic dipole interaction ($\mu m\mu$) as designated in Fig. 1c. Furthermore, a chiral medium with an applied static magnetic field not only breaks the parity-inversion and time-reversal symmetry simultaneously but also induces a new physical process, the MCh effect ($im m\mu$), as shown in Fig. 1d.

The overall MCh effect depends on the inner product of the magnetic field and the wavevector of the radiation together with the derivative chiral dispersion as phenomenologically described in the Methods section. We demonstrate a strong ExMCh effect in a two-dimensional TB-WS$_2$ on a TIG substrate. Figure 1e

shows the schematic of the demonstration, in which the TIG provides a spontaneous magnetic moment for a magnetic proximity effect at room temperature. Meanwhile, the TB-WS$_2$ with a twisting angle of $\theta$ offers a strong chirality that leads to a net population difference between the two valley excitons, which we observe via luminescence spectra. Moreover, optically excited valley excitons in WS$_2$ monolayers enhance the chirality via spin–orbit coupling and magnetic responses via magnetic exchange interactions simultaneously and hence demonstrating strong ExMCh effect. The optical image in Fig. 1f shows the TB-WS$_2$ we exfoliate on the ferromagnetic TIG substrate with the top monolayer twisted clockwise by an angle of 26°. We determine the relative stacking angles by comparing the microscopic edges since they tend to align well with crystallography orientations during the exfoliation[27,28].

**Excitonic chirality with nonmagnetic substrates**. We obtain a geometrically induced chirality that corresponds to the scenario in Fig. 1b by stacking two ML-WS$_2$ with a twisting angle of $\theta$ on an oxidized silicon substrate that is nonmagnetic (Fig. 2a). We understand the chirality by the overlapping, or so-called dephasing[29], of the separated wave functions at the middle point of the top and bottom monolayers[30]. Without time-reversal symmetry breaking, the two wave functions in real-space have opposite phase but with the same periodicity of $2\pi/6$ because of the hexagonal lattice of the monolayers. As a result, the chirality reaches the maximum with the overlapping of the two wave functions being the largest by shifting at the twisting angle around $\pi/6$, in consistent with that in twisted-graphene layers[31]. More importantly, the optically excited valley excitons enhance the geometrically induced chirality in TB-WS$_2$ because of the strong spin-orbit coupling[32]. Equivalently, Fig. 2b shows that the shifting of wavefunctions induced by chirality leads to an equal population difference but with opposite signs ($V+$ and $V-$) determined by the relative twisting directions (clockwise or counterclockwise) for the two valleys located at the K-point of the first Brillouin zone[33]. Details of the optical chirality in twisted bilayers are in the Supplementary Information.

During the experiment, a linearly polarized light at the wavelength of 560 nm impinges on the WS$_2$ and excites the valley excitons. We subsequently analyze the left and right circular polarization (LCP and RCP) components of the photoluminescence emission. Governed by the optical selection rules, those circular components lock with the two valley excitons ($\sigma^-$ and $\sigma^+$) accordingly. By twisting the top monolayer counterclockwise with an angle of 28°, we observe a higher luminescence intensity for $\sigma^-$ excitons (solid blue) compared to that for $\sigma^+$ excitons (solid red) in Fig. 2c, signifying the existence of the geometrically induced chirality. The amplitude of the difference of excitonic photoluminescence reaches a value of ~5%, equivalent to rotate the polarization plane of ~2.6°. We further verify the existence of the chirality by twisting the top monolayer clockwise with an angle of 26° (Supplementary Fig. 1). Indeed, we observe the chirality flips resulting in a lower luminescence intensity for $\sigma^-$ excitons (solid blue) under the same excitation condition. The absolute amplitude of chirality, however, is difficult to compare owing to unknown interlayer slip as described in the Supplementary Information. On the other hand, with an ML-WS$_2$ on an oxidized silicon substrate, Fig. 2d shows an identical behavior between the two photoluminescence spectra. This phenomenon is because the linearly polarized light excites an equal amount of valley excitons in the ML-WS$_2$. Without the geometrically induced chirality, the population difference between $\sigma^-$ and $\sigma^+$ vanishes consequently. The result also unambiguously shows that our experiments have corrected

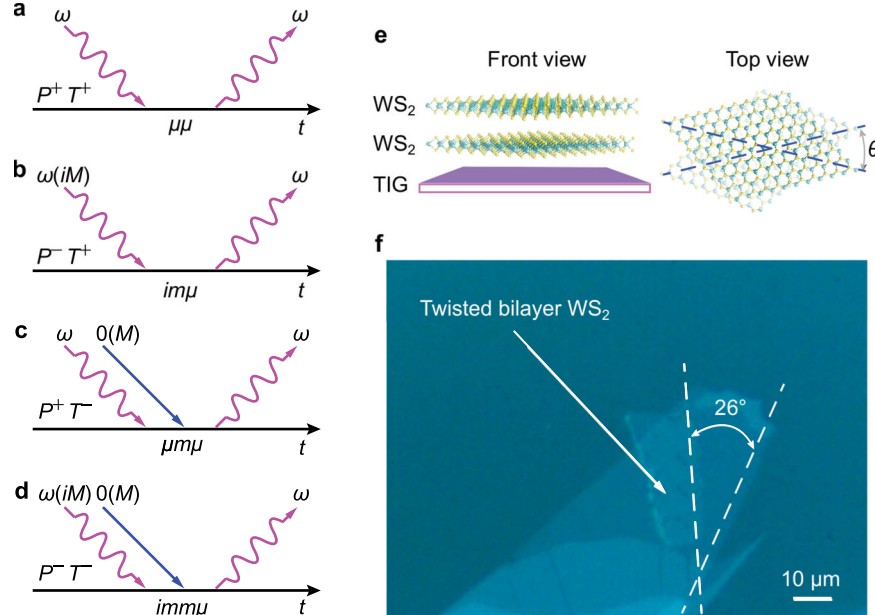

**Fig. 1 Light-matter interactions in chiral media. a–d** Multipolar light-matter interactions with symmetry analysis. The horizontal axis is the time ($t$), and the arrows are light waves (purple) at the frequency of $\omega$ and a static magnetic field (blue), correspondingly. In a comment medium **a** electric dipole interactions ($\mu\mu$) generally dominate light-matter interaction processes, in which the parity-inversion ($P$) and time-reversal symmetry ($T$) are both conserved ($+$). The breaking of parity-inversion symmetry ($P^-$) in a chiral material (**b**) leads to magnetic dipoles ($m$) interacting with electric dipoles ($\mu$) while keeping the time-reversal symmetry conserved ($im$). An external magnetic field breaks the time-reversal symmetry and introduces magnetic dipoles in a general material. **c** By breaking both the two symmetries simultaneously ($P^-T^-$), a second-order product ($imm\mu$), or so-called magneto-chiral (MCh) effect, stands out in a chiral medium in the presence of a static magnetic field (**d**). **e** Schematic shows that stacking two tungsten disulfide monolayers with a twist ($\theta$) offers a chirality and the ferromagnetic thulium iron garnet (TIG) provides an exchange magnetic field. **f** Optical images of a twisted bilayer tungsten disulfide (TB-WS$_2$) with the twisting angle of 26° on the TIG. The scale bar represents a length of 10 μm.

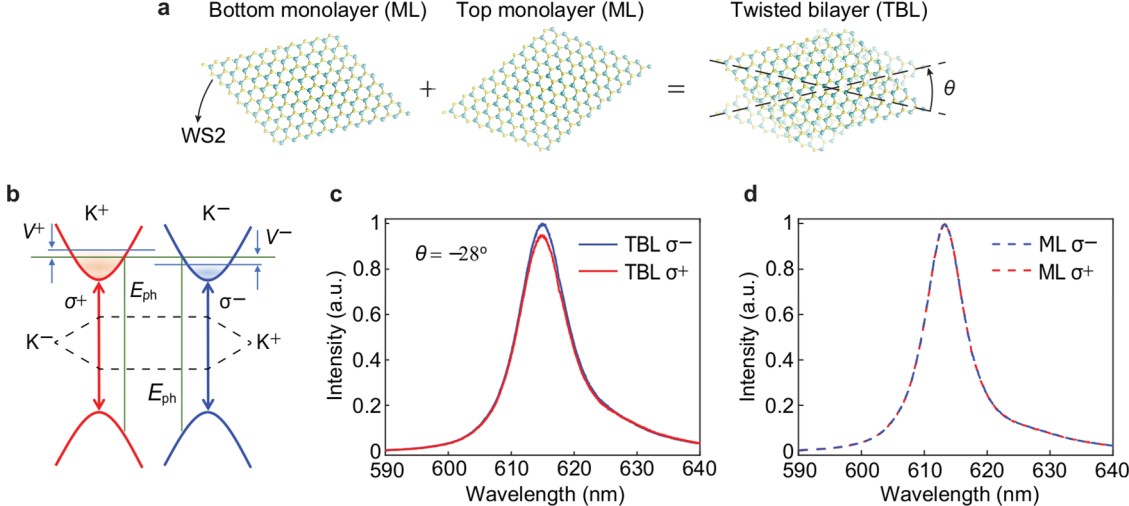

**Fig. 2 Excitonic chirality in twisted WS$_2$ monolayers on an oxidized silicon substrate. a** Twisting two WS$_2$ monolayers with a deterministic angle of $\theta$ breaks the parity-inversion symmetry, which leads to a geometrically induced chirality (Fig. 1b), interacting distinctly with left- and right-circularly polarized light (LCP and RCP), in the homostructure. **b** The handiness of the input light (LCP and RCP) locks with the helicity of the photoluminescence for the two valleys ($\sigma^-$ and $\sigma^+$) at the K-point of the Brillouin zone. The twisting of monolayers equivalently exerts a population difference to the two valleys ($V+$ and $V-$), leading to the intensity difference of the photoluminescence between them. **c** A twisted bilayer (TB) WS$_2$ possesses geometrically induced chirality, and hence the luminescence intensity for $\sigma^-$ and $\sigma^+$ excitons are different (solid curves). The top monolayer twists counterclockwise with an angle of 28° leading to a higher luminescence intensity of $\sigma^-$ exciton (solid blue) compared to that of $\sigma^+$ exciton (solid red). The amplitude of the difference in luminescence intensity reaches a value of ~5%. **d** A linearly polarized light that comprises an equal amount of LCP and RCP shows a vanishing preference on the excitation of the two valleys of $\sigma^-$ and $\sigma^+$ in a monolayer (ML) WS$_2$ on a SiO$_2$/Si substrate at room temperature (dashed curves). Which unambiguously shows we have corrected all system errors that might lead to any discrepancy of the two valleys, verifying the geometrically induced chirality in **c**.

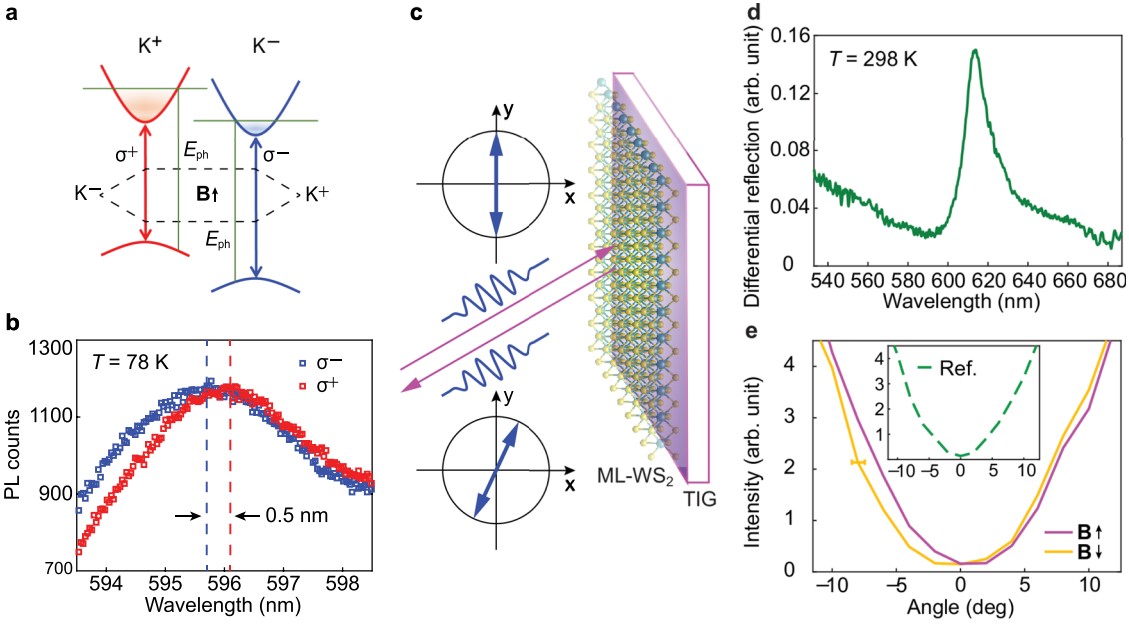

**Fig. 3 Magneto-optical effects in a monolayer WS₂ on TIG. a** The time-reversal symmetry breaking with an out-of-plane exchange magnetic field provided by TIG induces a spectral splitting and an intensity difference between $\sigma^-$ (blue) and $\sigma^+$ (red) in the ML-WS₂ under the pumping of $E_{ph}$. **b** The observed splitting of ~0.5 nm (~1.7 meV) corresponds to an exchange magnetic field of ~7 T at the temperature of 78 K from the TIG. The splitting is estimated to be ~0.2 nm that is hardly measurable in spectra with the reduced exchange magnetic field of ~2.7 T at 298 K. **c** An alternative method to characterize the magneto-optical effect (Fig. 1C) is using the rotation of linear polarization (blue arrows) of light upon reflection. **d** A differential reflection spectrum, defined as the difference in reflectance of the sample (WS₂) and substrate (TIG) normalized to that of a reference (silver mirror), consists of a peak of light-matter interaction at the wavelength of ~617 nm. **e** Analysis of the polarization of reflected light from WS₂ at 617 nm with up (purple) and down (yellow) magnetic moments. The 0° represents that the polarization of the output and input light is the same. The minima of curves designate the polarization of the outgoing light showing a rotation angle of ~2° compared to that of the input light. By flipping the exchange magnetic field (**B ↑** and **B ↓**), the rotation angle changes the sign indicating that the rotation is due to the magneto-optical effect. In the absence of the exchange magnetic field, the polarization of the light reflected from a reference silver mirror (inset) is the same as that of the input light, which manifests as the minimum at 0°, further confirming the magneto effect being the origin of the rotation. The error bar of ±0.5° is system uncertainty.

all system errors that could lead to the luminescence difference between the two valley excitons.

**Magneto-optical effects in a monolayer atomic crystal.** With time-reversal symmetry breaking, the valley excitons also facilitate the coupling with an out-of-plane magnetic field that lifts the degeneracy of the two valleys leading to a spectral splitting as well as a population difference between $\sigma^-$ (blue) and $\sigma^+$ (red) excitons in the ML-WS₂ under the pumping of $E_{ph}$, where $E_{ph}$ is the pumping energy (Fig. 3a). Instead of externally applying a strong magnetic field, we introduce a spontaneous magnetic moment from a ferromagnetic substrate (TIG) to boost such magnetic responses via magnetic exchange interactions between WS₂ and TIG. According to the magnetic response in Supplementary Fig. 2, we magnetize the TIG using a permeant magnet with a small magnetic field (~100 Oe). At the temperature of 78 K, we observe a splitting of ~0.5 nm (~1.7 meV) between the two valleys using linearly polarized excitation light at the wavelength of 560 nm (Fig. 3b). The observed splitting ($\Delta E_{K-K'}$) effectively corresponds to an exchange magnetic field ($B_{eff}$) of ~7 T given that $\Delta E_{K-K'} = 4\mu_B B_{eff}$, where $\mu_B$ is the Bohr magneton[21]. The $B_{eff}$ further reduces to ~2.7 T at the room temperature governed by Bloch's law for the temperature dependence of spontaneous ferromagnetism, providing that the Curie temperature of TIG is ~400 K[26]. The first-order approximation of Bloch's law is the following: $M(T) = M(0)\left[1 - \left(\frac{T}{T_C}\right)^{1.5}\right]$, where $M(0)$ is the spontaneous magnetization at absolute zero, $T$ is temperature, and $T_C$ is Curie temperature. Therefore, the splitting is further estimated

to be ~0.2 nm that is hardly observable in excitonic photoluminescence spectra with thermal broadening at room temperature.

Fortunately, a magnetically induced optical activity with a static magnetic field as that in Fig. 1c also rotates the plane of linear polarization at room temperature. We manage to measure the magnetically induced rotation of an ML-WS₂ using a differential reflection technique with a broadband white-light source under the exchange magnetic field provided by the TIG (Fig. 3c). This differential reflection technique with the experimental setup shown in Supplementary Fig. 3 is similar to the so-called polar magneto-optical Kerr effect (MOKE) measurements commonly used for studying magneto-optical materials[34–36]. In particular, the differential reflection refers to the difference in reflectance between a sample (ML-WS₂) and substrate (TIG) normalized to that of a reference (silver mirror). Based on a scattering process, this technique overcomes the difficulty caused by the dephasing of valley photoluminescence at the same time reflects the absorption of the material[37]. A typical differential reflection spectrum for the ML-WS₂ encompasses a significant peak of light-matter interaction located at the wavelength of ~617 nm (Fig. 3d). Compared to that at 78 K (Fig. 3b), the increase of the peak wavelength, or the decrease of the photon energy, is due to the thermal reduction of the electronic bandgap[38]. At the peak wavelength, we detect the magnetically induced rotation by analyzing the polarization of the reflected light compared to that of the incident light (Fig. 3e).

To increase the sensitivity, we place the sample in between a pair of polarizers with the optical axis of the polarizer at the output terminal nearly perpendicular to that of the polarizer at

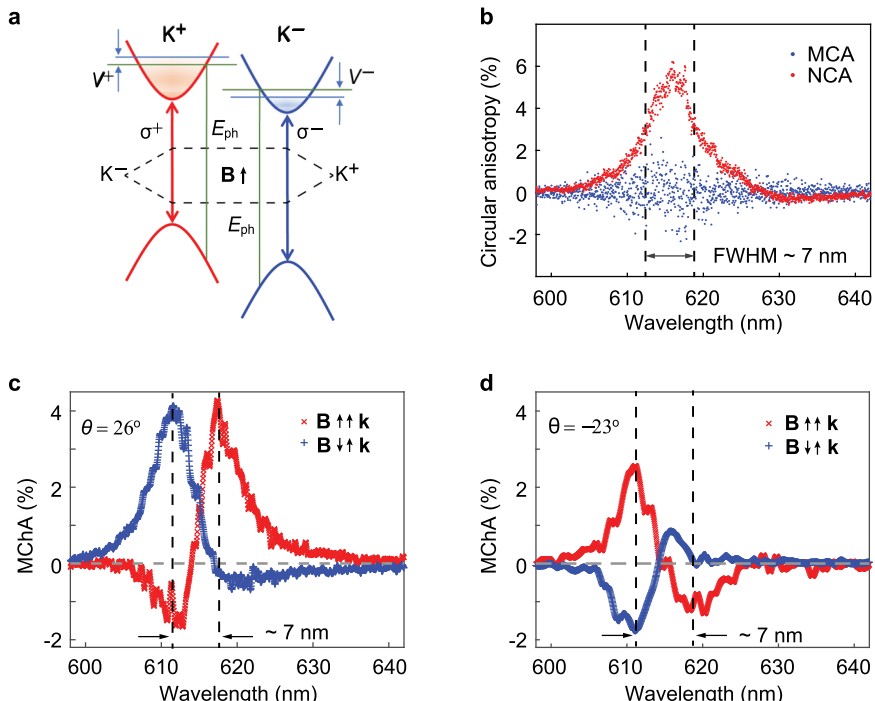

**Fig. 4 The strong excitonic magneto-chiral anisotropy. a** The schematic of the magneto-chiral effect that lifts the degeneracy and exerts a population difference to the two valleys simultaneously. **b** Circular anisotropy spectra for $\sigma^-$ and $\sigma^+$ excitons at room temperature. The natural circular anisotropy (NCA) spectrum (red) extracted from the same sample in Fig. 2c that situated on an oxidized silicon substrate possesses a significant peak with the full width at half maximum (FWHM) of ~7 nm, signifying the chirality. The circular anisotropy spectrum (blue) induced by the spontaneous magnetic field (MCA) for an ML-WS$_2$ on TIG is zero at all wavelengths. The magneto-chiral anisotropy (MChA) spectra for a twisting angle of 26° (**c**) and −23° (**d**) show opposite spectral behaviors, except both of them, contain a peak and a dip with a spectral distance precisely the same as the FWHM of NCA in **b**. Moreover, for both **c** and **d**, when the spontaneous magnetic field flip from parallel (red cross) to antiparallel (blue plus) to the wavevector **k**, the spectra also flip.

the input terminal (see details in the Methods). In this case, the minimum, ideally zero, output intensity designates that the polarization of light right behind the reflection plane is precisely perpendicular to the output polarizer. The location of the minima referenced to that of the input light, which we assign the angle of 0°, is the angle of rotation caused the exchange magnetic field. The sign of the minima is the direction of the magnetically induced rotation, either clockwise (+) or counterclockwise (−). With the ML-WS$_2$ on TIG, Fig. 3e clearly shows a rotation of ~2° that results from the strong exchange magnetic field provided by the ferromagnetic substrate at room temperature. By flipping the magnetic field, the sign of the rotation is also flipped, confirming that the rotation is due to the magnetic field of TIG. Indeed, without the magnetic field, light reflected from a silver mirror shows zero rotation of the polarization (Fig. 3e, inset). Although a polar MOKE measurement procedure at room temperature can resolve better than $\pm 0.02 \times 10^{-3}$ degrees[39], our resolution is around ±0.1 degrees, mainly limited by the uncertainty of reading angles on the mechanical rotational mount for polarizers. However, this resolution is sufficient for unambiguously measuring the polarization rotation of ~2 degrees in our experiments.

**Observation of the excitonic magneto-chiral effect.** We have distinctly observed the natural (chirality) and magnetically induced circular anisotropy (MCA), all evident in magnetic dipole interactions in two-dimensional atomic layers. While the first one is due to the parity-inversion symmetry breaking, the second is because of the time-reversal symmetry breaking. Here we find a link, the ExMCh effect, that unifies the two phenomena by stacking two ML-WS$_2$ with a twist on the ferromagnetic TIG, which breaks both the parity-inversion and time-reversal

symmetry simultaneously as shown in Fig. 4a. The ExMCh effect is not a simple linear superposition of the chirality and magnetically induced optical activity but rather a second-order effect. Mathematically, the ExMCh effect is proportional to the inner product of the magnetic field and wavevector together with the derivative of the chirality (see details in the Methods). Phenomenologically, the spectral splitting in the ExMCh effect equals to the full width at half maximum (FWHM) of the spectrum for natural circular anisotropy (NCA), which is in distinct contrast to the splitting for the MCA.

The circular anisotropy is the difference in photoluminescence intensity for the two valley excitons normalized to the average of the two. We obtain the NCA spectrum from the TB-WS$_2$ on an oxidized silicon and MCA spectrum from the ML-WS$_2$ on TIG. Figure 4b shows that the FWHM of NCA (red) is ~7 nm, and the spectral splitting of MCA (blue) is not observable at 298 K. For a TB-WS$_2$ with a twisting angle of 26° (clockwise) under an estimated exchange magnetic field of ~2.7 T, we observe a peak and a dip in the magneto-chiral anisotropy (MChA) spectra in Fig. 4c. For a single-peak spectrum, its derivative spectral behavior will be a peak and a dip instead. Therefore, our observation validates the theoretical analysis (see Methods section) that the ExMCh effect is related to the derivative of single-peak chirality in Fig. 4b. We obtain the MChA with a differential PL spectrum by subtracting the two PL spectra for the two valley excitons, because the two valleys lock with each other by time-reversal, equivalent to the reversal of the wavevector (**k**)[32,37]. The spectral distance of ~7 nm is precisely the FWHM of NCA for chirality, and it is around 35 times larger than the magnetically induced splitting estimated at room temperature (~0.2 nm). Moreover, we achieve a flip of the spectral behavior for

the ExMCh effect by flipping the magnetic moment of TIG, which changes the exchange magnetic field from parallel to antiparallel to the wavevector of the luminescence. This observation verifies the second characteristic feature that the pseudoscalar of **B·k** determines the ExMCh effect. The MChA that designates the ability to generate enantiomeric excess reaches a value of ~4% at room temperature. Furthermore, when the twisting angle changes from clockwise (26°) to counter-clockwise (−23°), the MCh behaviors also flip, as shown in Fig. 4d. The results confirm the observation of the ExMCh effect in the twisted atomic layers.

In summary, we observe an ExMCh effect in twisted bilayer WS₂ induced by a spontaneous magnetic moment of the ferromagnetic substrate (TIG) at room temperature. In the observed ExMCh effect, the twisting of two atomic layers artificially breaks the parity-inversion symmetry, and the ferromagnetism spontaneously breaks the time-reversal symmetry. More importantly, the chirality, magnetism, and valley excitons manifest in a unified manner, which could provide a new way to manipulate the valley degree of freedom for valleytronics[40,41]. Furthermore, the electronic nature of valley excitons fascinates the manipulation of enantiomeric excess in MCh systems for photochemical reactions[17], molecular synthesis[42], and drug delivery[43].

## Methods

**Interpretation of the MCh effect**. A phenomenological description of electro- and magneto-optical effects is the expansion of the natural dielectric susceptibility, $\varepsilon_N$, with vectors and tensors that characterize either the wave itself or an external force[44]. With the terms related to this work, the expansion is the following: $\varepsilon_\pm(\omega, \mathbf{B}, \mathbf{k}) = \varepsilon_N(\omega) \pm \varepsilon_M(\mathbf{B}) \pm \varepsilon_{Ch}(\mathbf{k}) + \varepsilon_{MCh}(\mathbf{B}, \mathbf{k})$, where $\omega$ is the angular frequency of the right (+) and left (−) circularly polarized light, **B** is a magnetic field, and **k** is a wavevector; the subscription of M, Ch, and MCh are the magnetically induced rotation, chirality, and MCh effect, respectively. For a general isotropic medium, the well-known magnetically induced rotation, Faraday or magnetic Kerr effect, results from a Larmor precession of electrons with the angular velocity $\Omega_L = -e\mathbf{B}/2mc$, where $e$ is the negative electron charge, $m$ is the mass, and $c$ is the velocity of light in vacuum. Therefore, the right and left circularly polarized light experiences the dielectric susceptibility, not at the frequency of $\omega$ but a shifted frequency of $\omega_\pm = \omega \pm \Omega_L$. And hence, the dielectric susceptibility for the magneto-optical effect writes as:

$$\varepsilon_\pm(\omega \pm \Omega_L) = \varepsilon_N(\omega) \pm \frac{|e|}{2mc} \cdot \frac{\partial \varepsilon_N}{\partial \omega} \mathbf{B}. \tag{1}$$

On the other hand, the dielectric susceptibility of a chiral medium is described by $\varepsilon_\pm(\omega) = \varepsilon_N(\omega) \pm \alpha_{Ch}(\omega)\mathbf{k}$. Similarly, light waves in a chiral medium under an externally applied magnetic field experience a new dielectric susceptibility:

$$\varepsilon_\pm(\omega \pm \Omega_L) = \varepsilon_N(\omega \pm \Omega_L) \pm \alpha_{Ch}(\omega \pm \Omega_L)\mathbf{k}, \tag{2}$$

which is also written as

$$\varepsilon_\pm(\omega, \mathbf{B}, \mathbf{k}) = \varepsilon_N \pm \frac{|e|}{2mc} \cdot \frac{\partial \varepsilon_N}{\partial \omega} \mathbf{B} \pm \alpha_{Ch}\mathbf{k} + \frac{|e|}{2mc} \cdot \frac{\partial \alpha_{Ch}}{\partial \omega}(\mathbf{B} \cdot \mathbf{k}), \tag{3}$$

where the last term signifies the MCh effect that depends on the inner product of the magnetic field and wavevector together with the derivative of the chirality-induced dielectric susceptibility. An oversimplified graphical illustration of the phenomenological model is in the Supplementary Information (Supplementary Fig. 4).

**Device fabrication**. A 10-nm thick TIG (Tm₃Fe₅O₁₂) film was grown on a single-crystal substituted gadolinium gallium garnet (SGGG) substrate via a pulsed-laser deposition at the temperature of ~500 °C with oxygen and ozone in the chamber[26]. Because of the large lattice-mismatch between SGGG and TIG as well as the negative magnetostriction constant, the TIG film provides an intrinsic exchange magnetic field that is mainly in the out-of-plane direction for WS₂ atomic layers in the close vicinity. The vibrating sample magnetometry characterized ferromagnetic responses of the TIG films at room temperature. The WS₂ monolayers were mechanically exfoliated from a bulk crystal using tape and subsequently transferred onto target samples, such as the TIG and a silicon wafer with an oxide layer with the thickness of 285 nm on top. The WS₂ monolayers were identified using optical contrast and photoluminescence spectroscopy (Supplementary Fig. 5). For twisted bilayer devices, the first WS₂ monolayer situated on the TIG that mounted on a rotational stage, and the second WS₂ monolayer stacked onto the first one with an angle by rotating the stage with an accuracy of ~1° (Supplementary Fig. 6).

**Photoluminescence spectroscopy**. A WS₂ atomic layer changes from an indirect to direct-bandgap semiconductor when thinning down to the monolayer limit, which features with a significantly large photoluminescence peak at the wavelength of ~615 nm at room temperature. In sharp contrast to a natural bilayer WS₂, a TB-WS₂ preserves the direct-bandgap nature by stacking two monolayers with a twist. Two sets of photoluminescence spectroscopy were employed to characterize the devices. The first one was performed to identify WS₂ monolayers using a Raman system (Horiba Labram HR Evolution) under normal incidence of linearly polarized excitation light at the wavelength of 473 nm. A ×100 objective collected the photoluminescence of the sample with an integration time of 5 s. A charge-coupled device camera after a grating of 600 g/mm recorded the spectral behavior of the photoluminescence. The second photoluminescence spectroscopy was performed with ultrafast excitation laser pulses at the wavelength of 560 nm to characterize the devices. The time-averaged power of the laser pulses that have a duration of 100 fs and a repetition rate of 80 MHz is ~145 µw. After passing through a set of polarizers and waveplates for controlling the intensity and polarization, the laser pulses impinge on and excite the sample with the photoluminescence collected by a ×40 objective. The excited photoluminescence was separated from the excitation laser by a set of filters and analyzed with a quarter waveplate and linear polarizer together with a grating of 1200 g/mm before being collected by an electronically cooled silicon camera. During the experiment, all spectra using the latter photoluminescence spectroscopy are an average of five consecutive measurements.

**Differential reflection spectroscopy**. The two valleys at the K-point of the first Brillouin zone excited by LCP and RCP light quickly lose the coherency at room temperature, which is the so-called dephasing effect that causes a randomized polarization in photoluminescence spectra. Consequently, the photoluminescence spectroscopy hardly detects the angle of rotation induced by the magnetically induced effect with linearly polarized incident light under an external magnetic field. However, the differential spectroscopy overcomes the difficulty caused by the dephasing effect at the same time reflects the absorption of the material by detecting the difference in reflectance between a sample and substrate normalized to that of a reference with white light. The linearly polarized white light is from a fiber-coupled tungsten halogen source with the polarization being further tailored by a half-wave plate before impinging on the sample through a ×40 objective. A set of half-wave plates and linear polarizers in front of the spectrometer analyze the polarization of light reflected from the sample (Supplementary Fig. 3). The difference between the two polarizations before and after reflecting from the sample is the angle of the magnetically induced rotation caused by the magnetic field.

## Data availability

All data supporting this study and its findings are available within the article and its Supplementary Information or from the corresponding author upon reasonable request.

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

## Acknowledgements

This work was supported by the Gordon and Betty Moore Foundation (award no.5722) and the Ernest S. Kuh Endowed Chair Professorship. S.L. acknowledges the start-up funding from Texas A&M University and the Governor's University Research Initiative (GURI).

## Author contributions

S.L. and X.Z. conceived the idea and initiated the project. S.L. formulated the theory. S.L., X.L., H.Z., C.G., and S.Y. built the experimental setup and carried out the optical measurements. S.L. and S.W. fabricated the devices. Y.L and J.S. provided the ferromagnetic substrates and carried out the magnetic characterizations. S.L., Y.W., and X.Z. analyzed the data and wrote the manuscript with input from all authors. X.Z. and Y.W. supervised the work.

## Competing interests

The authors declare no competing interests.
