## [Peer Review File · Nature Communications]

Reviewers' Comments:

Reviewer #1:

Remarks to the Author:

The manuscript shows interesting results, which they interpreted as the magneto-chiral and excitonic magneto-chiral effects by atomic thin twisted-bilayer WS₂ supported on a ferromagnetic substrate, thulium iron garnet (TIG). In my opinion, the results' interpretation is not well scientifically supported by how it is presented. Additionally, it is not clear if the magnitude of the observed phenomena is or not within the range at which it should be. For example, in figure 1 the differences in the magnitude can be estimated from the different interactions between electric and magnetic dipoles.

Finally, my main concern is with the experiment that is based on reflectance measurements. Here, during the reflection, a natural inversion occurs. How is this taken into account in the interpretation of the result? Is it possible to measure chiral effects in such arrangements? First, a parity inversion occurs during the reflection; and second, reflectance measurements loss resolution on the wavevector phase. From my perspective, this experimental configuration is not possible to measure this kind of effects, unless something else is done to break the symmetry again. It is not clear if this is done or not.

Additionally, how much is the reflectance resolution measurements to be sure that the difference can be attributed to the phenomena and not the apparatus? For example, differential reflectance has been used to measure atomic changes in surface, but under ultra-high vacuum conditions, large crystalline surfaces, etc. In this way, it is possible to measure such differences as those reported here. However, it is not clear that the small differences observed here are not from humidity adsorption or other experimental conditions that have not been controlled.

Reviewer #2:

Remarks to the Author:

RE NCOMMS-20-25795: Observation of strong excitonic magneto-chiral anisotropy in twisted bilayer van der Waals crystals

In this work, Lan et al. demonstrate a link between geometrically induced chirality and magnetically induced circular anisotropy. They use a twisted bilayer of WS₂ on top of the ferromagnetic Thulium Iron Garnet (TIG) substrate that breaks both parity-inversion and time-reversal symmetry. They show the geometrical induced chirality by stacking two monolayers of WS₂ at a particular angle; they observe a change in PL intensity for the two components of σ^+ and σ^- . For the magnetically induced circular anisotropy, they use a WS₂ monolayer and observe a splitting of the two circular components.

It is a very nice and systematic work in a very rapidly evolving field of the 2D materials beyond graphene. The timing of this work is excellent for the large community working on this field. The important result of this work is summarized in Fig.4 where a strong excitonic magneto-chiral anisotropy was demonstrated.

There are some points I would like the authors to elaborate more:

1. How reproducible are the results?
2. The authors show data from one or two twisted bilayer (TBs) samples with a twist angle of 28 or 26 degrees. Did they perform a systematic study of the magneto-chiral anisotropy as function of the twisted angle? It will make a very strong case a study with TBs where they vary the angle around 30 degrees with the step of 1 degree.
3. At several parts of the paper, it will be very helpful for the reader to see how the experiment was done. For example, see Figure 2c and 2d. What is the polarization state of the excitation laser? Is it linearly polarized and then the PL signal was analyzed σ^+ and σ^- ?

4. Figure 3b: Where the PL spectra taken under zero magnetic field or did the authors have applied a small magnetic field to magnetize the ferromagnetic substrate? Please add few sentences to make this part much clearer.
5. How the stacking angle was determined? Did the authors perform any second harmonic generation measurements? The authors should discuss this part in more detail.
6. Fig. 4c and 4d. How much is the applied magnetic field? As in my point 3 above, in many parts of the paper, the authors should give more details about the conditions the spectra were recorded.
7. Figure 4c. The authors claim that the difference between the peak and the dip matches exactly the FWHM in Figure 4b and that verifies their theoretical prediction. What is the prediction? Is there any theory support for that?
8. Comparison between Fig 4c and Fig 4d: Despite the fact that the twist angle is 26 and -23 degrees (i.e. not exactly mirror image), the energy difference between the peak and dip position is the same, 7nm. Why is that? Does this mean the twist angle plays no role, or that this energy difference has a large error bar?
9. Let's stay on Fig.4c. The twist angle is 26 degrees. The authors should show data also from another structure where the twist angle is different but still positive, for example 22, 24 or 28 degrees.

In conclusion, this is a nice work and it can contribute to the rapidly evolving field of the 2D materials. Nevertheless, there are many points to be taken into account before any further consideration.

Response to Reviewer 1's comments: NCOMMS-20-25795

We appreciate the time for **Reviewer 1** to review our manuscript (NCOMMS-20-25795) and for offering critical comments. While acknowledging our results being “*interesting*,” the **Reviewer** had several technical comments/questions mainly focused on reflectance measurements. We clarify that the reflectance measurements use a so-called polar magneto-optical Kerr effect (MOKE) setup, which is indeed capable of measuring our findings of polarization rotation induced by an out-of-plane exchange magnetic field from the ferromagnetic substrate. We have added the discussions on the reflectance measurements in the revised manuscript (pages 7 and 8). We are grateful for the **Reviewer**'s insightful comments that help our revised manuscript being even more technically sound.

We have taken the **Reviewer**'s comments under serious consideration and addressed all the points of concern. In the following part, we present a point-by-point response to the **Reviewer**'s comments quoted using *italics*.

The manuscript shows interesting results, which they interpreted as the magneto-chiral and excitonic magneto-chiral effects by atomic thin twisted-bilayer WS₂ supported on a ferromagnetic substrate, thulium iron garnet (TIG). In my opinion, the results' interpretation is not well scientifically supported by how it is presented. Additionally, it is not clear if the magnitude of the observed phenomena is or not within the range at which it should be. For example, in figure 1 the differences in the magnitude can be estimated from the different interactions between electric and magnetic dipoles.

We thank the **Reviewer** for recognizing the results as being "interesting." While respecting other opinions, we clarify that Figure 1 with the Hamiltonian described with electric and magnetic dipoles are for qualitatively understanding of light-matter interactions. For quantitatively estimating the strength of those interactions, particularly in 2D materials, it demands a more sophisticated model in a microscopic picture that is out of the scope of this work. However, we can evaluate the strengths by comparing them with theoretical or experimental results in the literature. For the chirality induced by twisting two atomic semiconductor layers (for example, WS₂ in our work), the theoretical investigation in Ref. 19 [*Nanophotonics* **7**, 753-762 (2018)] shows that the circular polarization of the photoluminescence emission ranges from 0.2% to 15%. Note that their definition of circular polarization different from ours by a factor of two. Therefore, our result of 5% in Figure 2c well situates within the range. For the magneto-optical effect, the experimental results in Ref. 25 [*Nature Nanotechnology* **12**, 757-762 (2017)] shows an enhanced valley energy splitting of ~2.5 meV with an estimated exchange magnetic field of ~12

T from a ferromagnetic substrate at the temperature of 7 K. Our result in Figure 3b shows a comparable valley splitting of ~ 1.7 meV with an exchange magnetic field of ~ 7 T at the temperature of 78 K. For the magneto-chiral effect in 2D materials, to the best of our knowledge, there is no similar scenario in the literature. However, using compound molecules in bulk, a magneto-chiral anisotropy of $\sim 1.2\%$ is recently obtained in Ref. 9. Our observation of 4% is more than three times larger, which we believe is due to the coupling with excitons in the 2D materials. Therefore, we clarify that all our results situate well within the range they should be.

Finally, my main concern is with the experiment that is based on reflectance measurements. Here, during the reflection, a natural inversion occurs. How is this taken into account in the interpretation of the result? Is it possible to measure chiral effects in such arrangements? First, a parity inversion occurs during the reflection; and second, reflectance measurements loss resolution on the wavevector phase. From my perspective, this experimental configuration is not possible to measure this kind of effects, unless something else is done to break the symmetry again. It is not clear if this is done or not.

We agree with the **Reviewer** that a parity inversion does occur during the reflection, and it loses the resolution in the phase without extra symmetry breaking. Therefore, the reflection setup cannot distinguish the first-order optical responses from the left and right circularly polarized (LCP and RCP) light. In our measurement, this scenario is precisely the control experiment without a magnetic field (Figure 3e, inset), manifesting with a minimum difference at 0 degrees. However, our objective using the reflectance measurements is to measure the polarization rotation induced by the exchange magnetic field from the ferromagnetic substrate (Figure 3e). With this exchange magnetic field, we can measure the polarization rotation accordingly, as shown in Figure 3e, with the minima of difference deviate from the 0 degrees. These measurements agree well with the **Reviewer's** argument of symmetry breaking, considering that the exchange magnetic field breaks the symmetry automatically. We are grateful to the **Reviewer** for this insightful question/comment, which in turn verifies our work being technically sound. In fact, this reflectance measurement of magnetically induced polarization rotation is also the so-called polar magneto-optical Kerr effect (MOKE) commonly used for studying magneto-optical materials [Review of Scientific Instruments **85**, 103702 (2014), Applied Physics Letters **109**, 122406 (2016), and Nano Letters **20**, 3435-3441 (2020)]. We have added those three references to the revised manuscript. Please note that some reference numbers could differ from the previous submission because of the new reference papers added to the revised manuscript. For a better understanding of the measurements, we also add more descriptions of the reflection technique. “This differential reflection technique with the

experimental setup shown in Figure S6 is similar to the so-called polar magneto-optical Kerr effect (MOKE) measurements commonly used for studying magneto-optical materials³⁴⁻³⁶.”

Additionally, how much is the reflectance resolution measurements to be sure that the difference can be attributed to the phenomena and not the apparatus? For example, differential reflectance has been used to measure atomic changes in surface, but under ultra-high vacuum conditions, large crystalline surfaces, etc. In this way, it is possible to measure such differences as those reported here. However, it is not clear that the small differences observed here are not from humidity adsorption or other experimental conditions that have not been controlled.

This is a good question. We have ruled out the possibilities of the environments or artifacts being the causes of the observed results, and the reflectance measurement setup is capable of measuring the observations. With a similar polar MOKE measurement procedure at room temperature, the resolution of the reflectance measurement setup is better than $\pm 0.02 \times 10^{-3}$ degrees [*Physical Review B* **96**, 235132 (2017)]. In our measurements, the resolution is around ± 0.1 degrees, mainly limited by the uncertainty of reading angles on the mechanical rotational mount for polarizers. This resolution, however, is more than sufficient for using the setup to measure the polarization rotation angle (~ 2 degrees) induced by the exchange magnetic field of ~ 2.7 T. We add the discussion of the resolution with the reference mentioned above to the revised manuscript as the following. “Although a polar MOKE measurement procedure at room temperature can resolve better than $\pm 0.02 \times 10^{-3}$ degrees³⁹, our resolution is around ± 0.1 degrees, mainly limited by the uncertainty of reading angles on the mechanical rotational mount for polarizers. However, this resolution is sufficient for unambiguously measuring the polarization rotation of ~ 2 degrees in our experiments.”

We then use three ways to verify that the polarization rotation indeed results from the ferromagnetic substrate. **1)** The polarization rotation from the control experiment without a magnetic field is 0 degrees (Figure 3e, inset), while that for the sample on the ferromagnetic substrate is ~ 2 degrees. This observation rules out the environment being the cause of the polarization rotation. **2)** By flipping the direction of the exchange magnetic field, we observe that the induced polarization rotation flips the sign (Figure 3e). This observation verifies that the polarization rotation does result from the exchange magnetic field of the ferromagnetic substrate. **3)** By comparing the polarization rotation from the sample on-resonance (617 nm) and off-resonance (650 nm) in Figure S8 in the Supplementary Information, we show that our observation stems from the excitonic effects from the sample.

We can further validate the results of reflectance measurements by comparing the polarization rotation in Figure 3e with the magnetically induced valley energy splitting in Figure 3b. A valley splitting of ~ 0.5 nm (~ 1.7 meV) corresponds to an effective exchange magnetic field of ~ 7 T at the temperature of 78 K, as shown in the manuscript. The effective magnetic field reduces to ~ 2.7 T at room temperature, governed by Bloch's law for the temperature dependence of spontaneous ferromagnetism. As a result, the valley splitting reduces to ~ 0.2 nm at room temperature as well. Meanwhile, the resonance of excitons has a spectral width of ~ 14 nm, corresponding to a π -phase shift. Therefore, the polarization rotation can be roughly estimated as ~ 2.6 degrees (i.e., $0.2 \text{ nm} / 14 \text{ nm} * 180 \text{ degrees} = 2.6 \text{ degrees}$). With the differential reflectance measurement setup, our measurement result is ~ 2 degrees, matching relatively well with the estimation from the photoluminescence spectroscopy measurement at the temperature of 78 K. Therefore, we are confident that our work is solid and technically sound.

Response to Reviewer 2's comments: NCOMMS-20-25795

We are very grateful to **Reviewer 2** for the overall positive assessment of our manuscript (NCOMMS-20-25795). The **Reviewer** evaluated it as "*a very nice and systematic work*" and "*the timing of this work is excellent,*" and concluded that "*it can contribute to the rapidly evolving field of the 2D materials.*" In the meantime, the **Reviewer** commented and suggested several occasions to show how we perform the experiments. Following the suggestions, we have added many details on the experiments and revised several descriptions in the revised manuscript. We have also clarified that the result of these experiments is not a single event nor an artifact. Those comments and suggestions help improve not only the technical content but also the presentation of the paper.

We have taken the **Reviewer's** comments under serious consideration and addressed all the points of concern. In the following part, we present a point-by-point response to the **Reviewer's** comments quoted using *italics*.

1. How reproducible are the results?

Our results in the manuscript are solidly reproducible. During the experiment, all photoluminescence spectra using the ultrafast pumping laser and detected by the electronic cooled silicon camera after the spectrograph are an average of five consecutive measurements. We have made sure the finding of the work is not a single event nor an artifact. **1**) For the chirality, other than using the sample with a counterclockwise twisting angle of 28° (Figure 2), we also have one with a clockwise 26° (Figure S1).

The chirality-induced valley polarization in both the two scenarios has a similar amplitude, $\sim 5\%$ for the former and $\sim 4\%$ for the latter. The discrepancy is mainly due to the angular difference and other uncertainties such as the interlayer slip, as shown in the Supplementary Information. **2)** For magnetically induced optical effects, we verified the results using two entirely different characterization methods (valley splitting at low temperature in Figure 3b and differential reflectance for polarization rotation at room temperature in Figure 3e). **3)** For the magneto-chiral effect, we also used two different samples with opposite twisting angles ($+26^\circ$ and -23°). And again, both the two cases support the magneto-chiral anisotropy with a characteristic feature that is spectrally derivative to the corresponding chiral spectrum. With repeated observations from different samples, our findings in the manuscript are thus reproducible. We add the description of the photoluminescence spectroscopy in the Methods section as the following. “During the experiment, all spectra using the latter photoluminescence spectroscopy are an average of five consecutive measurements.”

2. The authors show data from one or two twisted bilayer (TBs) samples with a twist angle of 28 or 26 degrees. Did they perform a systematic study of the magneto-chiral anisotropy as function of the twisted angle? It will make a very strong case a study with TBs where they vary the angle around 30 degrees with the step of 1 degree.

We thank the **Reviewer** for acknowledging that we have used a couple of samples with different twisting angles ($+26^\circ$ and -23°) to observe the magneto-chiral anisotropy (Figure 4c and 4d). As shown in the *Methods* section, a magneto-chiral effect holds three key aspects: the magnetic field, the wavevector, and the derivative of the chirality-induced dielectric susceptibility. **1)** The experiment with the two angles twisting in opposite directions is to induce opposite chirality. The opposite chirality further induces the flipping of the magneto-chiral anisotropy in Figures 4c and 4d. **2)** By flip the magnetic field (up and down), the magneto-chiral anisotropy achieves flip the spectral behavior on the same sample (red and blue). **3)** The two valleys have opposite momenta. By subtracting the two conjugate valley photoluminescence, the magneto-chiral anisotropy obtains the wavevector dependence. Therefore, our work has a complete data set for confirming the observation of the magneto-chiral effect. Although a detailed study with varying twisting angles around 30 degrees with the step of one degree may add more data, it is not absolutely necessary because confirmation of such a magneto-chiral effect was done by a complete set of data in verifying the three aspects of the chirality, magnetic field, and wavevector, as we discussed above.

3. At several parts of the paper, it will be very helpful for the reader to see how the experiment was done. For example, see Figure 2c and 2d. What is the polarization state of the excitation laser? Is it linearly polarized and then the PL signal was analyzed σ^+ and σ^- ?

We agree with the **Reviewer** on how we perform the experiments is crucial to the readers and the work. Following this valuable suggestion, we have added a substantial amount of experimental details to the revised manuscript. Specifically, in the experiments for Figures 2c and 2d, we used a linearly polarized light to excite the two valleys in the WS₂. We subsequently analyzed the σ^- and σ^+ components of the photoluminescence signals at the detection end. We have added the descriptions in the revised manuscript as the following. “During the experiment, a linearly polarized light at the wavelength of 560 nm impinges on the WS₂ and excites the valley excitons. We subsequently analyze the left and right circular polarization (LCP and RCP) components of the photoluminescence emission. Governed by the optical selection rules, those circular components lock with the two valley excitons (σ^- and σ^+) accordingly.” We also revised the description of Figure 2d in the main text. “On the other hand, with an ML-WS₂ on an oxidized silicon substrate, Figure 2d shows an identical behavior between the two photoluminescence spectra. This phenomenon is because the linearly polarized light excites an equal amount of valley excitons in the ML-WS₂. Without the geometrically induced chirality, the population difference between σ^- and σ^+ vanishes consequently.”

4. Figure 3b: Where the PL spectra taken under zero magnetic field or did the authors have applied a small magnetic field to magnetize the ferromagnetic substrate? Please add few sentences to make this part much clearer.

We highly appreciate the **Reviewer** for this insightful comment on the technical details. In the experiment for Figure 3b, we magnetized the ferromagnetic substrate by applying a small magnetic field (~100 Oe) from a permanent magnet. This small magnetic field is strong enough to magnetize the substrate, according to the magnetization response in Figure S3. Following the **Reviewer**'s suggestion, we add a description of the experiment for Figure 3b as the following. “According to the magnetic response in Figure S3, we magnetize the TIG using a permanent magnet with a small magnetic field (~100 Oe).”

5. How the stacking angle was determined? Did the authors perform any second harmonic generation measurements? The authors should discuss this part in more detail.

Instead of performing second-harmonic generation (SHG) measurements, we determine the stacking angle by comparing the microscopic edges. Though not as accurate as SHG for determining the crystalline structures, this method works comparably well for the chirality since it has a less stringent angular requirement than some phenomena such as the magic angle moiré physics. Guo and the colleagues' study [ACS Nano 10, 8980-8988, 2016] on the cleavage behaviors of two-dimension layered materials show that the in-plane crystallography orientations tend to align well with fracture edges. We also used transmission, reflection, and absorption measurements to confirm this tendency in previous work [ACS Photonics 3, 1176-1181, 2016]. Following the suggestion of the **Reviewer**, we add the details to the revised manuscript: " We determine the relative stacking angles by comparing the microscopic edges since they tend to align well with crystallography orientations during the exfoliation^{27,28}." We also insert the two papers into the reference list. Please note that the order for the rest references could differ from the previous version.

6. Fig. 4c and 4d. How much is the applied magnetic field? As in my point 3 above, in many parts of the paper, the authors should give more details about the conditions the spectra were recorded.

The exchange magnetic field is around 2.7 Tesla at room temperature for Figures 4c and 4d, estimated from the data in Figure 3b. At the temperature of 78 K, the valley splitting is 0.5 nm (~1.7 meV), corresponding to an exchange magnetic field (B_{eff}) of ~7 Tesla. The B_{eff} further reduces to ~2.7 Tesla at room temperature, governed by the Bloch's law for spontaneous ferromagnetism. We add more details for the estimation of exchange magnetic field: "The observed splitting ($\Delta E_{K-K'}$) effectively corresponds to an exchange magnetic field (B_{eff}) of ~7 T given that $\Delta E_{K-K'} = 4\mu_B B_{\text{eff}}$, where μ_B is the Bohr magneton²¹. The B_{eff} further reduces to ~2.7 T at the room temperature governed by Bloch's law for the temperature dependence of spontaneous ferromagnetism, providing that the Curie temperature of TIG is ~400 K²⁶. The first-order approximation of Bloch's law is the following: $M(T) = M(0)[1 - (\frac{T}{T_C})^{1.5}]$, where $M(0)$ is the spontaneous magnetization at absolute zero, T is temperature, and T_C is Curie temperature."

7. Figure 4c. The authors claim that the difference between the peak and the dip matches exactly the FWHM in Figure 4b and that verifies their theoretical prediction. What is the prediction? Is there any theory support for that?

We interpret the magneto-chiral effect by expanding the dielectric susceptibility, as shown in the *Methods* section. The theoretical analysis, though rudimentary, shows that the part in the dielectric susceptibility

function induced by the magneto-chiral effect is proportional to the magnetic field, wavevector, and the derivative of the chirality-induced dielectric susceptibility. For the last term, the derivative function corresponds to the slope of a curve. Because a single-peak spectrum has both negative and positive slopes, its derivative spectral behavior should show a peak and a dip instead. The chirality in Figure 4b (red) shows a spectral peak at the wavelength of ~ 617 nm. Therefore, we anticipate that the spectrum for the magneto-chiral effect will follow the derivative spectral behavior. Together with the response to Comment 6, we add the description for Figure 4c as the following. "For a TB-WS₂ with a twisting angle of 26° (clockwise) under an estimated exchange magnetic field of ~ 2.7 T, we observe a peak and a dip in the magneto-chiral anisotropy (MChA) spectra in Figure 4c. For a single peak spectrum, its derivative spectral behavior will be a peak and a dip instead. Therefore, our observation validates the theoretical analysis (see Methods section) that the ExMCh effect is related to the derivative of single-peak chirality in Figure 4b."

8. Comparison between Fig 4c and Fig 4d: Despite the fact that the twist angle is 26 and -23 degrees (i.e. not exactly mirror image), the energy difference between the peak and dip position is the same, 7nm. Why is that? Does this mean the twist angle plays no role, or that this energy difference has a large error bar?

We appreciate the **Reviewer** for this comment. The two cases ($+26^\circ$ and -23°) show the magneto-chiral anisotropy in the twisted atomic layers under an exchange magnetic field. From our theoretical analysis in the *Methods* section, the magneto-chiral effect has a derivative mathematical relation with the chirality. Accordingly, the peak and dip are a result of this derivative relationship. Therefore, their positions should not change for any cases. The two chiral samples with the twisting angles of $+26^\circ$ and -23° correspond to the two enantiomers or counterparts. As a result, their chirality, as well as the magneto-chiral effect, change the sign accordingly. For example, when the exchange magnetic field faces up (red curves), the peak in Figure 4c changes to a dip in Figure 4d at the same wavelength. As the **Reviewer** notifies, the two samples are not exactly mirror images. This angular difference induces a discrepancy in the amplitude of the magneto-chiral anisotropy. For $+26^\circ$, the amplitude is $\sim 4\%$, while for -23° is $\sim 3\%$. In other words, the roles that the twist angle plays lie in the sign and amplitude of the chirality and the magneto-chiral anisotropy.

9. Let's stay on Fig.4c. The twist angle is 26 degrees. The authors should show data also from another structure where the twist angle is different but still positive, for example 22, 24 or 28 degrees.

This comment is essentially the same as Comment 2. Referring to the previous response, we believe the data in our manuscript is a complete set that fully supports our claim of observing the excitonic magneto-chiral effect. We want to thank the **Reviewer** again for this comment, and it is inspiring for future research to study the details and evolution of this effect.

Reviewers' Comments:

Reviewer #1:

Remarks to the Author:

Authors answered satisfactorily the questions. In my opinion the manuscript can be accepted

Reviewer #2:

Remarks to the Author:

In the revised version of the paper, the authors have addressed all my concerns giving a detailed and satisfactory response to every point I made. My suggestions were taken into account and I believe that several sections of the manuscript are now improved and better presented.

Response to Reviewers' comments

We are glad that both reviewers are fully satisfied with our responses.

Reviewer #1:

Authors answered satisfactorily the questions. In my opinion the manuscript can be accepted.

We highly appreciate the **Reviewer** suggesting to accept our paper for publication.

Reviewer #2:

In the revised version of the paper, the authors have addressed all my concerns giving a detailed and satisfactory response to every point I made. My suggestions were taken into account and I believe that several sections of the manuscript are now improved and better presented.

We thank the **Reviewer** for acknowledging that our response is detailed and satisfactory. The **Reviewer's** suggestions have indeed improved our manuscript.